# Knowledge, attitudes, and practices of organ, tissue, and cell donation in Nicaragua

Jasley Navarrete[1]*, Engel Niño[1], Luis Moreno[1], Indiana Lopez Bonilla[2], Marvin Gonzalez-Quiroz[3,4]

1 National Autonomous University of Nicaragua at León (UNAN-León), León, Nicaragua, 2 Wuqu' Kawoq Maya Health Alliance, Chimaltenango, Guatemala, 3 Department of Environmental and Occupational Health, The University of Texas School of Public Health San Antonio, The University of Texas Health Science Center at San Antonio, San Antonio, Texas, United States of America, 4 Department of Renal Medicine, University College London, London, United Kingdom

* arisleydi2009@yahoo.es

## Abstract

Organ donation and transplantation are essential for improving the quality of life of people with organ failure due to chronic diseases (e.g., chronic kidney disease) or irreparable organ damage from accidents. In Nicaragua, chronic kidney disease of unknown etiology (CKDu) has emerged as a significant public health challenge, disproportionally affecting young agricultural workers and leading to premature deaths. Despite enactment of Law 847 in 2013, which regulates organ donations and transplantation, Nicaragua faces critical challenges, including lack of awareness, inadequate infrastructure, and limited public dissemination on the value of organ donation leading to an increasing number of patients on waiting lists. To address these gaps, we conducted an online cross-sectional survey to assess the knowledge, attitudes, and practices (KAP) regarding organ donation and transplantation in Nicaragua, a lower-middle-income country. We conducted an online cross-sectional survey among 4,407 Nicaraguan residents aged 18 and above from all 15 departments and two regions between November 2022 and February 2023. Most participants were women (60.3%), people aged 18-35 years (79.9%), and residents in urban areas (62.8%). The findings revealed that only 28.6% had good knowledge regarding organ and tissue donation and transplantation, 91.9% expressed positive attitudes toward organ and tissue donation, being willing to donate regardless of religious believes (88.5%) or personal health conditions (90.0%). About 72.6% engaged in donation-related behaviors. Men, those with higher education, and unemployed participants showed greater adherence to these practices. In summary, while Nicaraguans show positive attitudes towards organ and tissue donation and transplantation, significant knowledge and supply-demand gaps persist. Targeted educational campaigns and infrastructure development are urgently needed to address these gaps, enhance public awareness, and promote organ donation, particularly in the context of CKDu´s burden on public health.

## Introduction

Organ donation and transplantation are the ideal lifesaving treatment for patients with end-stage disease of the heart, lungs, liver, kidney or accident victims with irreparable organ

**Data availability statement:** The data supporting the findings of this study are available within the manuscript (S1 Data).

**Funding:** The authors received no specific funding for this work.

**Competing interests:** The authors declare no competing interests.

damage. These organ transplants significantly improve the quality of life for recipients and increase their chances of long-term survival[1]. However, the gap between the demand for organs and their availability remains large, resulting in growing waiting lists globally[2].

The kidneys, heart, and lungs are the organs most in demand around the globe. By 2020, 76,000 kidney transplants were performed, mostly in developed countries [3–5]. Despite this, the number of people who have access to transplantation is still insufficient, with 850 million people affected by chronic kidney disease (CKD), and a quarter of them needing a renal transplant to improve their lives [6]. This gap is particularly wide in low- and middle-income countries where the infrastructure and capacity for organ procurement and transplantation are limited. Despite these limitations, renal transplants in Latin America have increased by 2.9 per million population from 2010 to 2019 [7].

The leading countries in renal transplantation in Latin America are Brazil and Mexico, while in Central America are Costa Rica and Guatemala [3,8–10]. Nicaragua has conducted 149 kidney transplants over the past 18 years, including 42 in children, predominantly among male recipients (66.4%) [11]. However, this progress is still not as fast as needed due to the burden of disease affecting vulnerable populations in the country due to the high prevalence of Chronic kidney disease of unknown etiology (CKDu) in Nicaragua, affecting over 21,000 patients [12] who are under renal replacement therapy, mostly peritoneal dialysis and hemodialysis [11,13].

In 2013, the Nicaraguan government enacted  Law 847, the "*Law on the Donation and Transplantation of Organs, Tissues, and Cells in Nicaragua regulates both the donation and transplantation of organs, tissues, and cells from living and deceased individuals*," which regulates the donation and procurement of organs, tissues, and cells from both living donors and deceased individuals. The law establishes guidelines for their use in therapeutic, educational, and research purposes in humans [14]. This law was put into action following a pivotal incident involving a renal transplant where the donor passed away within 24 hours post-surgery. It aims to regulate organ transplants from both deceased and live donors in the country. Despite the clear benefits of kidney transplants, Nicaragua currently lacks essential infrastructure, including an organ bank, retrieval system, and comprehensive oversight for organ donation [11]. Central America, particularly Nicaragua, faces a lack of comprehensive studies examining knowledge, attitudes, and practices (KAP) regarding organ donation and transplantation. Recognizing the impact of CKDu on public health and the need for organ donation awareness, we conducted a study to assess the Nicaraguans' knowledge, attitudes, and practices regarding organ donation and transplant in Nicaragua.

## Methods

### Ethics statement

The Ethics Committee of the National Autonomous University of Nicaragua, Leon, approved the study protocol, procedures, questionnaire, and consent statement (Act #236, Año 2022). Participants provided consent before completing the self-administered questionnaire.

### Study design and population

We conducted an online, population-based survey across all fifteen departments and two regions of the country from November 4th, 2022, to February 4th, 2023. The survey was open to Nicaraguan residents over 18 years of age, who could complete the structured questionnaire anonymously. It was disseminated through social media platforms such as WhatsApp, Twitter, Facebook, and Instagram. To maximize reach, we employed a convenient and snowball sampling method, encouraging participants to share the survey to others within their networks. Additionally, we leveraged the researchers' professional and personal networks, engaged with community leaders,

and collaborated with news pages and social media influencers to further promote the survey. The procedures for answering, the voluntary nature of participation, and the anonymity declaration were explained at the top of the questionnaire first page as part of the informed consent.

## Sampling

Epi Info 7 software was used to calculate a sample size of 3,840, based on the assumption that 50% of the population (over 18 years old; estimated at 3.9 million) would have good knowledge, practices, and positive attitudes regarding organ donation. The calculation considered a 95% confidence interval, a 5% margin of error, and a design effect of 10%. To ensure validity of the online survey, the following inclusion criteria were applied: participants had to be Nicaraguan citizens residing in the country, aged 18 years or older, of any sex, who completed the questionnaire and voluntary consented to participate. To minimize non-responses, all questions were made mandatory. A minimum of 20 individuals who did not complete the questionnaire were excluded from the analysis.

## Questionnaire content and data collection

A structured questionnaire with closed-ended questions was developed through a comprehensive multi-step process. The initial drafted was self-designed, drawing from existing, validated questionnaires recognized for their high reliability and validity in the context of organ and kidney transplantation [15–17]. This draft was then adapted to the Nicaraguan population with inputs from three experts in epidemiology, health statistics, and psychology.

To assess face validity, 30 residents from the city of Leon participated in the evaluation process. This allowed us to identify any unclear or ambiguous items and implement necessary linguistic and cultural adaptations to ensure the questionnaire's appropriateness for the local context. Each question was carefully reviewed to guarantee clarity and comprehension, ensuring that all participants would understand the survey as intended. The validity and reliability of the knowledge, attitudes and practices questionnaire were assessed by calculating Cronbach's alpha coefficient of 0.85. These calculations demonstrated the instrument's internal consistency and robustness.

This self-designed online questionnaire (S1 Questionnaire) was divided into four sections: (1) the participants' demographic and socioeconomic characteristics: this section collected data on sex, age, education, marital status, place of residence, religion, and occupation. (2) Knowledge of organ donations: this section included 13 statements related to organ donation, with response options yielding a total score ranging from 0 to 28. Scores were categorized as follow: very good (>21 points), good (15-21 points), and deficient (≤14 points). (3) Attitudes toward organ donation: Participants' attitudes were measured using 14 items on a two-step Likert scale (Agree/Disagree). Scores ranged from 0 to 28 and were categorized as positive (>16 points) and negative (≤16 points). (4) Practices related to organ donation: This section included seven statements, presented as yes/no questions, with total scores ranging from 0 to 21. The scores were categorized as adequate (>12 points) and inadequate (≤12 points).

## Data analysis

Data were analyzed using the Statistical Package for the Social Sciences (SPSS), version 25 for Windows. Descriptive statistics including frequencies, percentages, and chi-square tests were used to summarize the data. Multinomial logistic regression was employed to calculate odds ratio (OR) and 95% confidence intervals (CI) to evaluate associations of KAP measures. Covariates included sex, age, marital status, education level, religion, residency, and occupation. Statistical significance level was defined as $p < 0.05$.

## Results

A total of 4,407 Nicaraguans aged 18 and older participated in the study. Of the participants, 60.3% were women, aged between 18 and 35 years (79.9%), single (77.2%), identified as Catholic (63.6%), urban residents (62.8%), and employed (52.1%). (Table 1)

Overall, 4,347 participants (98.6%) correctly understood the concept of organ and tissue donation, but 17.6% were aware of Nicaragua's Law 847 (Law on the Donation and Transplantation of Organs, Tissues, and Cells for Humans). Knowledge gaps were evident since only 24.4% of participants could correctly identify organs and tissues eligible for donation and 28.6% knew the legality of organ donation and transplantation in Nicaragua. Less than half (47.2%) believed that donating an organ while alive could limit their quality of life, and 45.2% were aware of the requirements for becoming an organ donor in the country. (S1 Table).

Attitudes toward organ donation were predominantly positive. Most participants (88.5%) stated they would donate even if their religion prohibited it, and 90.0% were willing to donate if near death. Almost all participants (98.2%) believed transplants improve quality of life, and 97.9% viewed donation as an act of love. Despite this positivity, 75.3% expressed concerns about medical neglect if they were known donors. (S2 Table).

In terms of practices, fewer than half (48.4%) have ever donated blood, but 76.4% were willing to donate an organ to a severely ill family member or friend, and 82.2% agreed to sign a consent form authorizing the donation of their organs upon death. Nearly all respondents

**Table 1. Characteristics of the study population.**

| Characteristics | n | % |
|---|---|---|
| **Sex** | | |
| Male | 1,749 | 39.7 |
| Female | 2,658 | 60.3 |
| **Age group** (in years) | | |
| 18-35 | 3,522 | 79.9 |
| 36-90 | 885 | 20.1 |
| **Marital status** | | |
| Single | 3,402 | 77.2 |
| Married | 1,005 | 22.8 |
| **Education level** | | |
| No school/Elementary school | 339 | 7.7 |
| High School/Technical | 759 | 17.2 |
| University/Professional | 3,309 | 75.1 |
| **Religion** | | |
| Catholic | 2,802 | 63.6 |
| Evangelical | 915 | 20.8 |
| Other | 141 | 3.2 |
| None | 549 | 12.5 |
| **Residency** | | |
| Urban | 2,769 | 62.8 |
| Rural | 1,638 | 37.2 |
| **Job** | | |
| Yes | 2,298 | 52.1 |
| No | 2,109 | 47.9 |
| **Total** | **4,407** | **100.0** |

(97.8%) would accept an organ transplant if needed, and 98.0% were willing to donate their organs. (S3 Table).

Significant variation in KAP regarding organ and tissue donation and transplantation across sociodemographic groups. Over a quarter (28.6%) demonstrated good or very good knowledge which was higher among women (32.5%), adolescents and young adults (32.5%), single individuals (32.6%), highly educated participants (34.4%), urban residents (37.6%), unemployed individuals (30.6%) and participants practicing other religions different to catholic (40.4%). Overall, positive attitudes toward organ donation were high (91.9%) and 72.6% reported adequate practices. Women and younger adults demonstrated strong attitudes (93.4% and 94.8%, respectively), while men reported a higher rate of adequate practice (78.2%). No difference was observed by age group. Positive attitudes and adequate practices were also more common among individuals with higher education, single status, no religious affiliation, urban residency, and unemployment. (Table 2)

Table 3 shows the association between sociodemographic characteristics and KAP regarding organ and tissue donations. Men are less likely to have good knowledge (ORadj: 0.8, 95% CI: 0.6-1.1), a positive attitude (ORadj: 0.9, 95% CI: 0.5-2.0) and adequate practices (OR: 0.6, 95% CI: 0.4-0.7) compared to women. Younger participants, those with higher education levels, and urban residents showed higher odd of good knowledge (ORadj: 2.2,

**Table 2. Level of knowledge, attitudes, and practices regarding organ and tissue donation and transplantation by sociodemographic characteristics.**

| Characteristics | Knowledge | | | | | | *p*-value | Attitudes | | | | | *p*-value | Practices | | | | | *p*-value |
|---|---|---|---|---|---|---|---|---|---|---|---|---|---|---|---|---|---|---|---|
| | Poor | | Good | | Very good | | | Positive | | Negative | | | | Adequate | | Inadequate | | | |
| | n | % | n | % | n | % | | n | % | n | % | | | n | % | n | % | | |
| **Sex** | | | | | | | <0.001 | | | | | <0.001 | | | | | | <0.001 | |
| Male | 1,353 | 77.4 | 357 | 20.4 | 39 | 2.2 | | 1,560 | 89.2 | 189 | 10.8 | | | 1,368 | 78.2 | 381 | 21.8 | | |
| Female | 1,794 | 67.5 | 783 | 29.5 | 81 | 3.0 | | 2,490 | 93.7 | 168 | 6.3 | | | 1,833 | 69.0 | 825 | 31.0 | | |
| **Age group** (in years) | | | | | | | <0.001 | | | | | <0.001 | | | | | | 0.945 | |
| 18-35 | 2,388 | 67.8 | 1,026 | 29.1 | 108 | 3.1 | | 3,339 | 94.8 | 183 | 5.2 | | | 2,559 | 72.7 | 963 | 27.3 | | |
| 36-90 | 759 | 85.8 | 114 | 12.9 | 12 | 1.4 | | 711 | 80.3 | 174 | 19.7 | | | 642 | 72.5 | 243 | 27.5 | | |
| **Marital status** | | | | | | | <0.001 | | | | | <0.001 | | | | | | <0.001 | |
| Single | 2,295 | 67.5 | 999 | 29.4 | 108 | 3.2 | | 3,168 | 93.1 | 234 | 6.9 | | | 2,586 | 76.0 | 816 | 24.0 | | |
| Married | 852 | 84.8 | 141 | 14.0 | 12 | 1.2 | | 882 | 87.8 | 123 | 12.2 | | | 615 | 61.2 | 390 | 38.8 | | |
| **Education level** | | | | | | | <0.001 | | | | | <0.001 | | | | | | <0.001 | |
| No school/Elementary | 330 | 97.3 | 9 | 2.7 | 0 | 0.0 | | 264 | 77.9 | 75 | 22.1 | | | 240 | 70.8 | 99 | 29.2 | | |
| High School/Technical | 645 | 85.0 | 105 | 13.8 | 9 | 1.2 | | 654 | 86.2 | 105 | 13.8 | | | 420 | 55.3 | 339 | 44.7 | | |
| University/Professional | 2,172 | 65.6 | 1,026 | 31.0 | 111 | 3.4 | | 3,132 | 94.7 | 177 | 5.3 | | | 2,541 | 76.8 | 768 | 23.2 | | |
| **Religion** | | | | | | | <0.001 | | | | | 0.013 | | | | | | 0.945 | |
| Catholic | 2,121 | 75.7 | 615 | 21.9 | 66 | 2.4 | | 2,580 | 92.1 | 222 | 7.9 | | | 2,028 | 72.4 | 774 | 27.6 | | |
| Evangelical | 591 | 64.6 | 294 | 32.1 | 30 | 3.3 | | 837 | 91.5 | 78 | 8.5 | | | 660 | 72.1 | 255 | 27.9 | | |
| Other | 84 | 59.6 | 45 | 31.9 | 12 | 8.5 | | 120 | 85.1 | 21 | 14.9 | | | 102 | 72.3 | 39 | 27.7 | | |
| None | 351 | 63.9 | 186 | 33.9 | 12 | 2.2 | | 513 | 93.4 | 36 | 6.6 | | | 411 | 74.9 | 138 | 25.1 | | |
| **Residency** | | | | | | | <0.001 | | | | | <0.001 | | | | | | 0.112 | |
| Urban | 1,728 | 62.4 | 936 | 33.8 | 105 | 3.8 | | 2,607 | 94.1 | 162 | 5.9 | | | 2,034 | 73.5 | 735 | 26.5 | | |
| Rural | 1,419 | 86.6 | 204 | 12.5 | 15 | 0.9 | | 1,443 | 88.1 | 195 | 11.9 | | | 1,167 | 71.2 | 471 | 28.8 | | |
| **Job** | | | | | | | 0.003 | | | | | < 0.001 | | | | | | <0.001 | |
| Yes | 1,683 | 73.2 | 546 | 23.8 | 69 | 3.0 | | 2,031 | 88.4 | 267 | 11.6 | | | 1,593 | 69.3 | 705 | 30.7 | | |
| No | 1,464 | 69.4 | 594 | 28.2 | 51 | 2.4 | | 2,019 | 95.7 | 90 | 4.3 | | | 1,608 | 76.2 | 501 | 23.8 | | |
| **Total** | **3,147** | **71.4** | **1140** | **25.9** | **120** | **2.7** | | **4,050** | **91.9** | **357** | **8.1** | | | **3,201** | **72.6** | **1,206** | **27.4** | | |

95% CI: 1.6-3.1) but significant differences in attitudes and practices toward organ donations. No differences in KAP were observed based on religion. Employment status increased the odds of having good knowledge (ORadj: 1.4, 95% CI: 1.0-1.8) and positive attitudes (ORadj: 2.2, 95% CI: 1.1-4.5), but showed no significant had impact on practices (ORadj: 1.0, 95% CI: 0.9-1.2). (Table 3)

## Discussion

This study reveals valuable insights into the factors influencing awareness and behaviors related to organ and tissue donation in the Nicaraguan Population. Despite nearly everyone were aware of organ donation, deep understanding remains insufficient, with near one-fourth accurately identifying organs and tissues eligible for donation and fewer being familiar with Nicaraguan's Law 847. Knowledge gaps were more pronounced among certain groups, including women, younger individuals, singles, and those with higher education. These disparities highlight the need for targeted educational initiatives.

While attitudes were very positive with 90% of participants willing to donate organs if near death and almost all viewing donation as an act of love. Three-quarters of respondents feared potential negligence if they were known donors. This mistrust, likely rooted in cultural

**Table 3. Association of sociodemographic characteristics with knowledge, attitudes, and practices regarding organ and tissue donation and transplantation.**

| Characteristics | Knowledge | | | | Attitudes | | Practices | |
|---|---|---|---|---|---|---|---|---|
| | Good | | Very Good | | Positive | | Adequate | |
| | OR (95%CI) | ORadj (95%CI) | OR (95%CI) | ORadj (95%CI) | OR (95%CI) | ORadj (95%CI) | OR (95%CI) | ORadj (95%CI) |
| **Sex** | | | | | | | | |
| Male | 0.6 (0.5-0.7) | 0.8 (0.6-1.1) | 0.6 (0.4-0.9) | 0.9 (0.5-2.0) | 0.6 (0.4-0.7) | 0.6 (0.5-0.8) | 1.6 (1.4-1.9) | 1.5 (1.3-1.8) |
| Female | 1.0 | 1.0 | 1.0 | 1.0 | 1.0 | 1.0 | 1.0 | 1.0 |
| **Age group** (in years) | | | | | | | | |
| 18–35 | 2.9 (2.3-3.5) | 1.3 (0.8-1.9) | 2.9 (1.6-5.2) | 1.3 (0.4-4.1) | 4.5 (3.6-5.6) | 2.6 (1.9-3.6) | 1.0 (0.8-1.2) | 0.5 (0.4-0.7) |
| 36–90 | 1.0 | 1.0 | 1.0 | 1.0 | 1.0 | 1.0 | 1.0 | 1.0 |
| **Marital status** | | | | | | | | |
| Single | 2.6 (2.2-3.2) | 1.8 (1.3-2.8) | 3.3 (1.8-6.1) | 2.8 (0.8-8.9) | 1.9 (1.5-2.4) | 0.7 (0.5-0.9) | 2.0 (1.7-2.3) | 1.7 (1.4-2.0) |
| Married | 1.0 | 1.0 | 1.0 | 1.0 | 1.0 | 1.0 | 1.0 | 1.0 |
| **Education level** | | | | | | | | |
| No school/Elementary | 0.1 (0.03-0.11) | 0.1 (0.04-0.4) | - (-) | - (-) | 0.2 (0.1-0.3) | 0.6 (0.4-0.9) | 0.7 (0.6-0.9) | 0.5 (0.4-0.7) |
| High School/Technical | 0.3 (0.2-0.4) | 0.5 (0.3-0.7) | 0.3 (0.1-0.5) | 0.4 (0.1-1.3) | 0.4 (0.3-0.5) | 0.6 (0.4-0.8) | 0.4 (0.3-0.4) | 0.4 (0.3-0.5) |
| University/Professional | 1.0 | 1.0 | 1.0 | 1.0 | 1.0 | 1.0 | 1.0 | 1.0 |
| **Religion** | | | | | | | | |
| Catholic | 0.5 (0.4-0.7) | 0.6 (0.4-0.9) | 0.9 (0.5-1.7) | 1.1 (0.4-3.4) | 0.8 (0.6-1.2) | 0.8 (0.5-1.2) | 0.9 (0.7-1.1) | 0.9 (0.7-1.1) |
| Evangelical | 0.9 (0.7-1.2) | 1 (0.7-1.6) | 1.5 (0.7-2.9) | 1.7 (0.5-5.8) | 0.8 (0.5-1.1) | 0.8 (0.5-1.2) | 0.9 (0.7-1.1) | 0.9 (0.7-1.2) |
| Other | 1.0 (0.7-1.5) | 0.9 (0.4-1.9) | 4.2 (1.8-9.6) | 3.9 (0.9-17.6) | 0.4 (0.2-0.7) | 0.3 (0.2-0.6) | 0.9 (0.6-1.3) | 0.9 (0.6-1.4) |
| None | 1.0 | 1.0 | 1.0 | 1.0 | 1.0 | 1.0 | 1.0 | 1.0 |
| **Residency** | | | | | | | | |
| Urban | 3.8 (3.2-4.4) | 2.2 (1.6-3.1) | 5.7 (3.3-9.9) | 3.2 (1.2-8.9) | 2.2 (1.7-2.7) | 1.2 (0.9-1.6) | 1.1 (0.9-1.3) | 0.9 (0.8-1.2) |
| Rural | 1.0 | 1.0 | 1.0 | 1.0 | 1.0 | 1.0 | 1.0 | 1.0 |
| **Job** | | | | | | | | |
| Yes | 0.8 (0.7-0.9) | 1.4 (1.0-1.8) | 1.2 (0.8-1.7) | 2.2 (1.1-4.5) | 0.3 (0.2-0.4) | 0.6 (0.4-0.8) | 0.7 (0.6-0.8) | 1.0 (0.9-1.2) |
| No | 1.0 | 1.0 | 1.0 | 1.0 | 1.0 | 1.0 | 1.0 | 1.0 |

Abbreviations: OR: Odd Ratio; 95%CI: 95% Confidence Interval; ORadj: Adjusted Odd Ratio.

beliefs [18,19], underscores the importance of fostering transparency, communication and trust in the healthcare institutions.[20,21] Tailored public awareness campaigns and policy adjustments could address these fears and encourage more widespread support for organ donation.[1,20] Behavioral practices reflect participants' knowledge and attitudes, with nearly three-quarters engaging in actions supportive of organ and tissue donations. However, fewer than half of the participants have ever donated blood. Statistically, urban living, higher education, and single status increase knowledge and positive attitudes.

Sociodemographic factors such as age, gender, education level, and socioeconomic status significantly influence KAP toward organ donation. A notable gender disparity emerges, with more females engaging in organ donation-related discussions and research. This reflects global trends where women are often more receptive to healthcare initiatives. In the Nicaraguan context, cultural factors, such as women's greater involvement in healthcare decision-making within families and communities, contribute to their higher participation rates. Additionally, women tend to be more active in seeking health-related information, making them more likely to support and participate in organ donation initiatives. In contrast, men, especially in rural areas, may be less involved in healthcare discussions, which could explain the lower participation rates among them.[22,23]

Law 847 – "*Law on the Donation and Transplantation of Organs, Tissues, and Cells in Nicaragua regulates both the donation and transplantation of organs, tissues, and cells from living and deceased individuals*" was introduced in 2013, representing a significant step toward regulating organ donation and transplantation in Nicaragua. Prior to this law, the country lacked comprehensive legal frameworks to govern organ donation, leading to inconsistent practices and limited public awareness. The survey highlighted a significant knowledge gap regarding the Law 847, stemming from insufficient public dissemination or the absence of an educational program to raise awareness among the population. Additionally, the country lacks an operational organ transplant program for both living or deceased donors. However, the survey also revealed that individuals with higher education levels tend to have greater knowledge and more likely to seek out information on the benefits and procedures, resulting in more positive attitudes toward it.[20] This trend is particularly evident in urban areas, where access to educational resources is more readily available. Similar findings have been reported in southern India [24], Nigeria [25,26], Paskitan [27], Egypt[28] and Spain[21] where participants with higher education demonstrated better knowledge of the concept of organ donation and transplantation, consent processes, and the ethical issues surrounding organ donation. Studies suggest that targeted educational interventions could help bridge the knowledge gap, particularly among populations with lower educational levels. In some LMICs, awareness campaigns aimed at students, healthcare professionals, and community leaders have been successful in improving knowledge and attitudes about organ donation [21,24–28].

Studies have consistently shown that women generally have higher levels of positive attitudes toward organ donation compared to men, often due to greater healthcare engagement [29]. However, Nicaragua's trend may be stronger than that seen in some countries due to the social role of women in family health management, which increases their exposure to healthcare topics. Despite these trends, widespread misconceptions and lack of information remain substantial barriers to increasing organ donation rates, both in Nicaragua and globally [30–32]. Many individuals still believe that organ donation contradicts religious beliefs or fear that signing up as a donor could compromise their medical care [18,19]. These misconceptions are particularly prevalent in rural and underserved areas of Nicaragua, where education and access to healthcare information is limited. Therefore, addressing these barriers requires targeted interventions that not only provide education but also challenge cultural

misconceptions, particularly those surrounding the perceived conflict between organ dona-tion and religious or cultural beliefs.

Our findings show positive attitudes toward organ donation and transplantation; however, a significant proportion of participants expressed fears about medical neglect if were known as donors. This concern, rooted in mistrust and misinformation, aligns with findings in other international studies. Wakefield et al. (2020) demonstrate that individu-als with favorable attitudes are more likely to register as donors and inform their families about their decisions [33]. However, negative attitudes driven to fear and lack of trust can significantly hinder donation rates. Trust in the healthcare system emerge as a crucial role; transparent communication and trust-building measures have been shown to mitigate these concerns and foster positive attitudes toward organ donations [34,35]. This is consistent with research from Spain participants who trusted the healthcare system were more likely to support organ donation and less likely to have concerns about medical neglect [21,36,37]. Similar research by Irving et al. emphasizes that rust in medical professionals is a key deter-minant in encouraging organ donation [38].

Nicaragua currently lacks an active organ donation and transplant program, as well as a waitlist for organ donors or recipients. Despite this, the population shown a good practice toward organ donation, which is influenced by both knowledge and attitudes. However, even individuals with positive attitudes may not take the necessary steps to become donors due to a lack of encouragement. Studies have found that simplifying the registration process and inte-grating it into routine activities, such as renewing a driver's license, can significantly increase donor registration rates. Furthermore, addressing sociodemographic factors is imperative. Younger individuals, for example, often have different attitudes compared to older adults, and targeted interventions for different age groups can enhance effectiveness [39–42]. Also, DeGroot et al. highlighted that younger people respond positively to social media campaigns and peer-led education programs, suggesting that modern communication channels can be effectively utilized to promote organ donation [43].

## Strengths and limitations

The study offers a detailed examination of the variables affecting the knowledge, attitudes, and behaviors related to organ and tissue donation in Nicaragua. Firstly, the robustness of this study lies in its large sample size (4,407 participants) and representative demographic diver-sity, providing a comprehensive snapshot of the nation's perspectives on this critical health-care issue. Secondly, it provides valuable insights into the current state of public knowledge, attitudes, and practices concerning organ, tissue, and cell donation within the Nicaraguan population, a region where such data may be limited. Thirdly, the online survey format allows for a broad reach, enabling participation from diverse demographic groups across the country. Additionally, the use of standardized questionnaires ensures validity and reliability of ques-tionnaire and consistency in data collection, facilitating the comparison of results with similar studies in other regions or countries.

Several limitations in this type of online cross-sectional survey should be considered. Firstly, as a self-reported online survey, the data may be influenced by response biases, such as social desirability bias and recall bias. Second, the sample was disproportionately com-posed of females, younger individuals, and participants with higher education levels. This reflects demographic trends in Nicaragua, where urban areas account for 65% the population is predominantly young (with over 40% having completed at least secondary education), and women often assume caregiving roles, fostering greater interest in health-related issues, including organ donation. Many may have also experienced family situations related to

organ donation, potentially influencing their participation. Third, the online nature of the survey, inherently excludes individuals with limited internet access or lower familiarity with the subject, which may further skew the results toward more educated participants. Fourth, the survey's pre-specified questions constrain the scope of confounders considered, potentially overlooking other influential factors related to organ donation willingness. Finally, the cross-sectional nature of the study offers only a snapshot of the current situation, limiting the ability to assess trends or of changes over time.

## Conclusions

This study highlights the significant influence of sociodemographic factors on KAP towards organ and tissue donation in Nicaragua. The findings highlight the critical need for targeted educational campaigns and trust-building initiatives to address knowledge gaps and promote actual donation practices. Specific strategies should focus on dissipating misconceptions, fear, and mistrust, and leveraging effective communication addressing different demographics, particularly those with lower knowledge or higher concern. By implementing such interventions, we can significantly improve organ and tissue donation rates and save more lives through enhanced practices.

## Supporting information

**S1 Table. Questions about the knowledge of tissue and organ donation.**
(DOCX)

**S2 Table. Questions about the attitudes toward tissue and organ donation.**
(DOCX)

**S3 Table. Questions about the practices toward tissue and organ donation.**
(DOCX)

**S1 Questionnaire. Knowledge, Attitudes and Practices toward organ donation and transplantation.**
(DOCX)

**S1 Data. Knowledge, Attitudes and Practices toward organ donation dataset.**
(XLSX)

**S1 Checklist. Inclusivity in global research.**
(DOCX)

**S2 Checklist. Human Subjects Research Checklist.**
(DOCX)

## Acknowledgments

The authors thank the study participants for completing the online survey. We express our appreciation to Dra. Arlen Soto from the Research Center on Health, Work, and Environment (CISTA) for her review and linguistic adaptation of the questionnaire and Dra. Aurora Aragón for her feedback and comments.

## Author contributions

**Conceptualization:** Jasley Navarrete, Engel Niño, Luis Moreno, Marvin González-Quiroz.

**Data curation:** Jasley Navarrete, Engel Niño, Luis Moreno, Marvin González-Quiroz.

**Formal analysis:** Jasley Navarrete, Engel Niño, Luis Moreno, Indiana López Bonilla, Marvin González-Quiroz.

**Methodology:** Jasley Navarrete, Marvin González-Quiroz.

**Supervision:** Marvin González-Quiroz.

**Writing – original draft:** Jasley Navarrete.

**Writing – review & editing:** Jasley Navarrete, Engel Niño, Luis Moreno, Indiana López Bonilla, Marvin González-Quiroz.

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
