## [Decision Letter · Decision Letter 0]

9 Dec 2024

PGPH-D-24-02008

Knowledge, Attitudes, and Practices of Organ, Tissue, and Cell Donation in Nicaragua

Dear Dr. Navarrete,

Thank you for submitting your manuscript to PLOS Global Public Health. After careful consideration, we feel that it has merit but does not fully meet PLOS Global Public Health’s publication criteria as it currently stands. Therefore, we invite you to submit a revised version of the manuscript that addresses the points raised during the review process.

Two key themes identified by the reviewers require special attention. First, the manuscript would be more informative for international audiences if your approach, your findings, and the Nicaraguan context were better situated within the international context.Second, the level of evidence available to justify the validity of your survey and its reliability for the population who responded could be better characterized. This should include a justification of the use of your questionnaire considering the availability of validated instruments that have been used in other settings.

We look forward to receiving your revised manuscript.

Kind regards,

W. Alton Russell, PhD

Academic Editor

Journal Requirements:

1. Please provide additional details regarding participant consent. In the ethics statement in the Methods and online submission information, please ensure that you have specified (1) whether consent was informed and (2) what type you obtained (for instance, written or verbal, and if verbal, how it was documented and witnessed). If your study included minors, state whether you obtained consent from parents or guardians. If the need for consent was waived by the ethics committee, please include this information.

If you are reporting a retrospective study of medical records or archived samples, please ensure that you have discussed whether all data were fully anonymized before you accessed them and/or whether the IRB or ethics committee waived the requirement for informed consent. If patients provided informed written consent to have data from their medical records used in research, please include this information."""

2. Please include a complete copy of PLOS’ questionnaire on inclusivity in global research in your revised manuscript. Our policy for research in this area aims to improve transparency in the reporting of research performed outside of researchers’ own country or community. The policy applies to researchers who have travelled to a different country to conduct research, research with Indigenous populations or their lands, and research on cultural artefacts. The questionnaire can also be requested at the journal’s discretion for any other submissions, even if these conditions are not met.  Please find more information on the policy and a link to download a blank copy of the questionnaire here: https://journals.plos.org/globalpublichealth/s/best-practices-in-research-reporting. Please upload a completed version of your questionnaire as Supporting Information when you resubmit your manuscript.

Reviewers' comments:

Reviewer's Responses to Questions

**Comments to the Author**

1. Does this manuscript meet PLOS Global Public Health’s publication criteria ? Is the manuscript technically sound, and do the data support the conclusions? The manuscript must describe methodologically and ethically rigorous research with conclusions that are appropriately drawn based on the data presented.

Reviewer #1: Yes

Reviewer #2: Partly

2. Has the statistical analysis been performed appropriately and rigorously?

Reviewer #1: Yes

Reviewer #2: I don't know

3. Have the authors made all data underlying the findings in their manuscript fully available (please refer to the Data Availability Statement at the start of the manuscript PDF file)?

Reviewer #1: Yes

Reviewer #2: No

4. Is the manuscript presented in an intelligible fashion and written in standard English?

Reviewer #1: Yes

Reviewer #2: Yes

5. Review Comments to the Author

Reviewer #1: The abstract presents the information correctly. The introductory part is considered adequate and consistent with the objectives and supporting literature. Methodology. This section could be developed. For example, it would be interesting to know the procedure. As for the evaluation instrument used, it is not a validated or standardized questionnaire. These instruments are currently being validated, for example PCID-DTO-RIOS.

The instrument used for this study does not describe the psychometric characteristics of the instrument, for examplee, reliability and validity. The description of the results are clearly stated and coherent with the proposed objectives. Regarding the discussion and conclusions, perhaps the results obtained could be compared, to a greater extent, with the results of other similar studies carried out in other countries. For example, there are many studies carried out in Spain, as it is a leading donation country. This would be an enrichment in terms of the quality of the work presented.

In general, it is a correct work, although the different sections could be completed to increase its quality.

Reviewer #2: This is a paper examining knowledge, attitudes and practices of organ donation. particularly in a lower-middle-income country. This is a very nice paper and helps fill gaps in research on the subject in Nicaragua and the wider Central American region.

I wish the authors every success in progressing this research - I have some comments/edits which I hope are helpful.

The inclusion of 4,407 participants provides robust data for the study. The use of a standardized questionnaire, linguistic adaptation, and pilot testing enhances the reliability and validity of the findings. The study obtained ethical approval, and the informed consent process was well-described.

Below are some of the things I think could be strengthen or addressed:

The sample bias- mostly women, younger, educated etc…most eligible deceased donors are older than this sample and likely not represent the general population…it feels like a bit more needs to be said and done to address the actual make-up of the population to test these further.

There is nothing here about live donation. Was this explored?

Limited Contextualisation: While the results are well-detailed, the discussion could benefit from more contextual comparison with similar studies in other low- and middle-income countries or regions facing similar challenges, and put more into context with this population – i.e. the recommendations are very generic and there is no priority given or suggestions how things can be implemented in a lower-middle country which has highly limited resources? I do not disagree with the discussion but it is very generic – what is specific to this population and demographic – or is it that the quick wins seen in developed countries and services are what need to be adapted – if so what is the priority and how can we make room for these in a country with limited resources/funds.

The law while interesting comes out of the blude and it is not clear what the question or use of it is trying to achieve, i.e. it does not explore in depth how this law has been implemented or its impact on public attitudes and behaviours, much more needs to be made of this if it is to be included, when introduced, what was there before, how was it advertised, does it work, what is the evidence etc etc…is it having impact on trust positive/negitive etc etc.

I was surprised to see that many of the questions appear to have open free text responses – and are omitted? If this is the case, why so? Much more could be done in the analysis to see what people are actually saying and where their KAPs might be amenable to intervention to strengthen this work and suggest more specific strategies for this specific population.

I am unable to comment in detail on the statistics, but as alluded to, it feels like more nuanced work is needed to address the KAP needs – this is rather a starting point than an end.

It is very interesting to read about the numbers and disparities and scarcity of donation – and that males are mostly getting kidneys – women are overrepresented on the waitlist for kidneys in the UK.

Some comments/limitations on the questions observed:

Some of the questions may be too general or simplistic to gauge nuanced understanding (e.g., “What is organ and tissue donation?”), as is suggested in the first line of the discussion – especially since none of the free text responses were analysed.

"What are the requirements to be an organ donor in Nicaragua?" this would require specialist knowledge not known by any general public.

The two-step Likert scale (Agree/Disagree) restricts nuanced responses, many people are indifferent, confused, uncertain when it comes to organ donation.

While the questionnaire asks about willingness and consent, it does not explore the reasons behind the lack of action (e.g., "Why have you not communicated your desire to donate?").

Questions also assume an understanding of concepts like blood donation, which may not be universal or uniformly accessible in Nicaragua, at the same time a recommendation might be to increase blood donors thus leading to tailored comms about organ donation.

Thank you for the opportunity to review this paper. I found it very interesting to read about organ donation in low-middle income countries.

6. PLOS authors have the option to publish the peer review history of their article (what does this mean? ). If published, this will include your full peer review and any attached files.

**Do you want your identity to be public for this peer review?** For information about this choice, including consent withdrawal, please see our Privacy Policy .

Reviewer #1: No

Reviewer #2: No

---

## [Decision Letter · Decision Letter 1]

5 Feb 2025

Knowledge, Attitudes, and Practices of Organ, Tissue, and Cell Donation in Nicaragua

PGPH-D-24-02008R1

Dear Doctor Navarrete,

We are pleased to inform you that your manuscript 'Knowledge, Attitudes, and Practices of Organ, Tissue, and Cell Donation in Nicaragua' has been provisionally accepted for publication in PLOS Global Public Health.

Best regards,

W. Alton Russell, PhD

Academic Editor

Reviewer Comments (if any, and for reference):

Reviewer's Responses to Questions

**Comments to the Author**

1. If the authors have adequately addressed your comments raised in a previous round of review and you feel that this manuscript is now acceptable for publication, you may indicate that here to bypass the “Comments to the Author” section, enter your conflict of interest statement in the “Confidential to Editor” section, and submit your "Accept" recommendation.

Reviewer #2: All comments have been addressed

2. Does this manuscript meet PLOS Global Public Health’s publication criteria ? Is the manuscript technically sound, and do the data support the conclusions? The manuscript must describe methodologically and ethically rigorous research with conclusions that are appropriately drawn based on the data presented.

Reviewer #2: Yes

3. Has the statistical analysis been performed appropriately and rigorously?

Reviewer #2: I don't know

4. Have the authors made all data underlying the findings in their manuscript fully available (please refer to the Data Availability Statement at the start of the manuscript PDF file)?

Reviewer #2: Yes

5. Is the manuscript presented in an intelligible fashion and written in standard English?

Reviewer #2: Yes

6. Review Comments to the Author

Reviewer #2: Congratulations and good luck progressing organ donation research and interventions in Nicaragua

7. PLOS authors have the option to publish the peer review history of their article (what does this mean? ). If published, this will include your full peer review and any attached files.

**Do you want your identity to be public for this peer review?** For information about this choice, including consent withdrawal, please see our Privacy Policy .

Reviewer #2: No
